# BMJ Open | Access to weight reduction interventions for overweight and obese patients in UK primary care: population-based cohort study

Helen P Booth, A Toby Prevost, Martin C Gulliford

## ABSTRACT

**Objectives:** To investigate access to weight management interventions for overweight and obese patients in primary care.

**Setting:** UK primary care electronic health records.

**Participants:** A cohort of 91 413 overweight and obese patients aged 30–100 years was sampled from the Clinical Practice Research Datalink (CPRD). Patients with body mass index (BMI) values ≥25 kg/m² recorded between 2005 and 2012 were included. BMI values were categorised using WHO criteria.

**Interventions:** Interventions for body weight management, including advice, referrals and prescription of antiobesity drugs, were evaluated.

**Primary and secondary outcome measures:** The rate of body weight management interventions and time to intervention were the main outcomes.

**Results:** Data were analysed for 91 413 patients, mean age 56 years, including 55 094 (60%) overweight and 36 319 (40%) obese, including 4099 (5%) with morbid obesity. During the study period, 90% of overweight patients had no weight management intervention recorded. Intervention was more frequent among obese patients, but 59% of patients with morbid obesity had no intervention recorded. Rates of intervention increased with BMI category. In morbid obesity, rates of intervention per 1000 patient years were: advice, 60.2 (95% CI 51.8 to 70.4); referral, 75.7 (95% CI 69.5 to 82.6) and antiobesity drugs 89.9 (95% CI 85.0 to 95.2). Weight management interventions were more often accessed by women, older patients, those with comorbidity and those in deprivation. Follow-up of body weight subsequent to interventions was infrequent.

**Conclusions:** Limited evidence of weight management interventions in primary care electronic health records may result from poor recording of advice given, but may indicate a lack of patient access to appropriate body weight management interventions in primary care.

Department of Primary Care and Public Health Sciences, King's College London, London, UK

**Correspondence to**
Helen P Booth;
helen.booth@kcl.ac.uk

### Strengths and limitations of this study

- This study uses primary care electronic health records to investigate the use of weight management interventions in overweight and obese patients.
- Lifestyle advice was the most commonly used intervention in all but morbidly obese patients, where antiobesity drugs were more frequent.
- Patients had to have a body mass index value recorded to be included in the study.
- Clinicians may be giving advice for weight management but not recording it.

quarter of adults are obese and up to two-thirds are overweight.[2] Primary care represents an important setting in which obese patients may access weight loss interventions. The main strategies for treating obesity are provision of lifestyle advice, referral for weight management, prescription of antiobesity drugs and, in severe cases, referral for bariatric surgery. Clinical guidelines recommend a stepped approach to weight management depending on the severity of a patient's obesity and whether they have weight-related comorbidities, with more intensive interventions offered as appropriate. Interventions should be agreed between the clinician and patient, and offered in conjunction with long-term follow-up and continuing care.[3]

Several studies have evaluated the effectiveness of primary care interventions for weight management in obesity,[4–8] but few studies have evaluated how overweight and obese patients are managed in primary care. A survey using data collected in 2000/2001, before the UK national guidelines on obesity management were published,[9] found that a fifth of obese patients were offered dietary counselling, less than 5% a referral and 2% antiobesity medications over an 18-month period. There is no more recent information on the use of interventions for the management of obesity in primary care.

## BACKGROUND

Obesity is a leading cause of premature morbidity and mortality worldwide.[1] In the UK, a

Access to appropriate weight management interventions for overweight and obese patients in primary care is of increasing importance in the context of a national objective to establish a downward trend in obesity among UK adults by 2020.[10] This study aimed to evaluate access in terms of recording and utilisation of weight management interventions for overweight and obesity using primary care electronic health records. Interventions were classified as lifestyle advice, referrals for weight management and prescription of antiobesity drugs.

## METHODS
### Data source and cohort definition
A cohort of patients was selected from the Clinical Practice Research Datalink (CPRD), a database of longitudinal patient electronic medical records from UK primary care. CPRD is the largest primary care database in the world, and represents over 5% of the UK population with about 680 practices currently contributing research quality data.[11] The initial cohort was selected as part of a larger project, and comprised a random sample of approximately 300 000 patients who were registered for at least 12 months with a general practice contributing data to CPRD between 1 January 2005 and 30 April 2012. Equal numbers of men and women were drawn from each year of the study without replacement. Patients were selected who had a body mass index (BMI) record indicating overweight or obesity during the study period.

### Exposure and outcome definitions
BMI was categorised using the WHO categories: overweight (BMI 25–29.9 kg/m$^2$), obese (BMI 30–34.9 kg/m$^2$), severe obesity (BMI 35.0–39.9 kg/m$^2$), morbid obesity (BMI ≥40 kg/m$^2$). Medical diagnoses of obesity in the medical record were also noted based on the presence of diagnostic codes. Morbidity status was ascertained based on the presence of 11 common conditions associated with obesity, including: type 2 diabetes, coronary heart disease, stroke, depression, osteoarthritis, back pain, joint problems, cancer, gallbladder disease, asthma and sleep apnoea. Smoking status and socioeconomic deprivation were also included as exposures. Socioeconomic deprivation was classified into quintiles using the Index of Multiple Deprivation rank based on patient postcode. Data on deprivation were only available for patients registered at English practices.

Interventions for the management of body weight were identified using medical codes recorded in clinical and referral records, recorded health promotion advice, and prescriptions for antiobesity drugs. For analysis, weight management interventions were classified into lifestyle advice, referrals for weight management and prescription of antiobesity drugs. Advice included codes relating to dieting, exercise and weight loss. Relevant referrals included those to community and hospital dieticians, for exercise therapy and for weight management programmes. Prescriptions for three different antiobesity

drugs were included; two of these, sibutramine and rimonabant, have been removed from the UK market because of safety concerns.[12 13] However, these drugs were in use during the time period investigated and so have been considered in this analysis. The only antiobesity drug currently licensed in the UK, orlistat, was introduced in 1998 and has been available over the counter as well as by prescription since 2009.[14] Multiple prescriptions of antiobesity drugs were considered to be a part of the same course of treatment if there was less than 6 months between prescriptions.

### Analysis
Person time was analysed following the index date; the first BMI record for overweight or obesity after 1 January 2005. Patient baseline characteristics were tabulated. The proportion of patients who received weight management interventions over the study period was evaluated by BMI category. Time-to-event analysis was used to calculate the rate of intervention utilisation by BMI category and to investigate variables associated with intervention using a multiple-failure multivariable Cox proportional hazards regression model with unordered events.[15] Variation in the use of weight management interventions by GP practice was investigated by calculating the proportion of patients receiving any intervention in the year following the index date. These data were then presented as percentiles of the distribution for all practices. Change in weight from baseline after the implementation of each type of intervention was calculated for up to 5 years of follow-up.

## RESULTS
Of the 300 006 patients in the cohort, 134 697 (45%) had an eligible BMI record. After patients with BMIs lower than 25 kg/m$^2$ were removed, data were analysed for 91 413 patients, with mean age 56 years, including 55 094 (60%) overweight and 36 319 (40%) or obese, including 4099 (5%) with morbid obesity. Mean age in men and women was 56 years. Patient characteristics on entry to the study are presented in table 1. At the index date (date of the first relevant BMI record) most patients were overweight (63.9% of men and 56.2% of women); 2.9% of men and 6.3% of women were morbidly obese. A diagnostic code for obesity was recorded for 3.9% of male patients and 6.5% of female patients. A higher proportion of women were non-smokers, while men were more likely to be former or current smokers.

The majority of patients did not receive a weight management intervention during the study period. The proportion of patients by BMI category with each type of intervention recorded on their medical record is given in table 2. In patients with morbid obesity, 60.0% of men and 58.1% of women had no record of weight management during the 7 years of the study. In patients with non-severe obesity (BMI 30–34.9 kg/m$^2$) the figures were 84.2% and 80.2%, respectively. The proportion of

**Table 1** Characteristics of overweight and obese patients

| | Men (48 413) | Women (43 000) |
|---|---|---|
| Mean age (SD) | 55.6 (13.9) | 56.4 (15.0) |
| BMI category (kg/m$^2$) | | |
| Overweight (BMI 25–29.9) | 30 950 (63.9) | 24 144 (56.2) |
| Obese (BMI 30–34.9) | 12 711 (26.3) | 11 364 (26.4) |
| Severe obesity (BMI 35–39.9) | 3368 (7.0) | 4777 (11.1) |
| Morbid obesity (BMI ≥40) | 1384 (2.9) | 2715 (6.3) |
| Medical code for obesity | 1876 (3.9) | 2810 (6.5) |
| Number of morbidities | | |
| 0 | 14 810 (30.6) | 9635 (22.4) |
| 1 | 14 988 (31.0) | 11 919 (27.7) |
| 2 | 10 323 (21.3) | 10 237 (23.8) |
| 3 or more | 8292 (17.1) | 11 209 (26.1) |
| Smoking status | | |
| Non-smoker | 17 415 (36.0) | 20 602 (47.9) |
| Ex-smoker | 15 188 (31.4) | 9916 (23.1) |
| Current smoker | 9359 (19.3) | 7448 (17.3) |
| Missing smoking status | 6451 (13.3) | 5034 (11.7) |
| IMD quintile | | |
| 1—least deprived | 11 490 (23.7) | 9229 (21.5) |
| 2 | 10 850 (22.4) | 9275 (21.6) |
| 3 | 8858 (18.3) | 7896 (18.4) |
| 4 | 7859 (16.2) | 7413 (17.2) |
| 5—most deprived | 6310 (13.0) | 6304 (14.7) |
| Missing IMD | 3046 (6.3) | 2883 (6.7) |

Figures are frequencies (column per cent) unless stated otherwise.
BMI, body mass index; IMD, Index of Multiple Deprivation.

patients who received an intervention increased with each additional BMI category. Advice was the most commonly recorded intervention in overweight and obese patients and severely obese men. Drug interventions were the most frequently recorded intervention in morbidly obese patients and severely obese women.

Rates of intervention are presented in table 3. Overall, the recorded rates of intervention were highest for advice at 30.3 (95% CI 29.3 to 31.4) per 1000 person-years. The rate of each intervention type increased in higher BMI categories. The rate of advice was 22.6 (21.6 to 23.8) per 1000 in overweight patients, and highest at 60.2 (51.8 to 70.4) per 1000 in morbidly obese patients. In overweight patients, advice was the most commonly used intervention, whereas drug prescription was the most common in morbidly obese patients.

The multivariable analysis identified BMI category as the strongest predictor of weight loss intervention, with a HR of 1.68 (95% CI 1.60 to 1.76) for obesity and 3.67 (95% CI 3.45 to 3.92) for morbid obesity (table 4). Increasing age, type 2 diabetes and depression tended to be associated with receiving a weight loss intervention. Female gender, being a former smoker and socioeconomic deprivation were associated with treatment for overweight and obesity.

**Table 2** Patients who received, or did not receive, a weight management intervention over the study period by gender and body mass index (BMI) category

| BMI category | Total | Advice | Referral | Drugs | No treatment |
|---|---|---|---|---|---|
| Men | | | | | |
| Overweight (BMI 25–29.9) | 30 950 | 1805 (5.8) | 913 (2.9) | 86 (0.3) | 28 282 (91.4) |
| Obese (BMI 30–34.9) | 12 711 | 1129 (8.9) | 762 (6.0) | 313 (2.5) | 10 697 (84.2) |
| Severe obesity (BMI 35–39.9) | 3368 | 363 (10.8) | 349 (10.4) | 333 (9.9) | 2499 (74.2) |
| Morbid obesity (BMI ≥40) | 1384 | 168 (12.1) | 239 (17.3) | 322 (23.3) | 831 (60.0) |
| Women | | | | | |
| Overweight (BMI 25–29.9) | 24 144 | 1331 (5.5) | 762 (3.2) | 451 (1.9) | 21 794 (90.3) |
| Obese (BMI 30–34.9) | 11 364 | 925 (8.1) | 740 (6.5) | 889 (7.8) | 9116 (80.2) |
| Severe obesity (BMI 35–39.9) | 4777 | 462 (9.7) | 445 (9.3) | 671 (14.0) | 3460 (72.4) |
| Morbid obesity (BMI ≥40) | 2715 | 284 (10.5) | 479 (17.6) | 724 (26.7) | 1578 (58.1) |

Figures are frequencies (row per cent).

**Table 3**  Rate of obesity management intervention by body mass index (BMI) category (per 1000 patient years), based on records of advice, referral or obesity drug prescription

|  | Advice | Referral | Drugs |
|---|---|---|---|
| Overall rate | 30.3 (29.3 to 31.4) | 20.0 (19.3 to 20.8) | 13.9 (13.5 to 14.4) |
| Overweight (BMI 25–29.9) | 22.6 (21.6 to 23.8) | 11.2 (10.5 to 11.9) | 2.9 (2.6 to 3.2) |
| Obese (BMI 30–34.9) | 36.4 (34.4 to 38.6) | 23.7 (22.3 to 25.2) | 15.7 (14.7 to 16.7) |
| Severe obesity (BMI 35–39.9) | 47.2 (42.7 to 52.3) | 38.4 (35.4 to 41.7) | 41.5 (39.0 to 44.1) |
| Morbid obesity (BMI ≥40) | 60.2 (51.8 to 70.4) | 75.7 (69.5 to 82.6) | 89.9 (85.0 to 95.2) |

There was substantial variation between practices in the recording of obesity management interventions (see table 5). The median proportion of obese and overweight patients receiving a weight management intervention during the study was 12% (IQR 7–19). A maximum of 91% overweight or obese patients in a practice had an intervention recorded. Follow-up measurements of body weight after intervention were most frequent in patients who had a referral, with 34.1% of patients having a weight measurement in the first year. In contrast, 20.7% of patients had a follow-up weight measurement in the first year after advice and 24.3% after a drug prescription. No trend in weight change was observed in patients up to 5 years after any of the three intervention types investigated.

## DISCUSSION
### Summary

Analysis of primary care electronic health records reveals that the use of weight management interventions in primary care for the treatment of overweight and obesity were infrequent between 2005 and 2012. The likelihood of intervention was strongly associated with BMI category. However, 60% of men and 58% of women with morbid obesity did not have any record of receiving weight management in primary care, with higher proportions noted in lower BMI categories. Variation in obesity management between general practices was evident, with many practices not recording any intervention. These results might be a consequence of poor documentation of advice given, but

**Table 4**  Cox proportional hazards model investigating time to multiple weight management interventions after a record of overweight or obesity

|  | Patients receiving weight management intervention (n) | Total patients (N) | HR | 95% CI | p Value |
|---|---|---|---|---|---|
| Age (decades) | – | – | 1.42 | 1.27 to 1.58 | <0.001 |
| Age squared | – | – | 0.97 | 0.96 to 0.98 | <0.001 |
| Gender |  |  |  |  |  |
| Male | 6104 | 48 413 | 1.00 | – | – |
| Female | 7054 | 43 000 | 1.14 | 1.10 to 1.19 | <0.001 |
| BMI category* |  |  |  |  |  |
| Overweight (BMI 25–29.9) | 5019 | 50 075 | 1.00 | – | – |
| Obese (BMI 30–34.9) | 4263 | 19 812 | 1.68 | 1.60 to 1.76 | <0.001 |
| Severe obesity (BMI 35–39.9) | 2186 | 5959 | 2.36 | 2.23 to 2.50 | <0.001 |
| Morbid obesity (BMI ≥40) | 1690 | 2409 | 3.67 | 3.45 to 3.91 | <0.001 |
| Smoking status |  |  |  |  |  |
| Non-smoker | 5441 | 32 576 | 1.00 | – | – |
| Former smoker | 3962 | 24 142 | 1.11 | 1.06 to 1.16 | <0.001 |
| Current smoker | 2530 | 14 277 | 0.99 | 0.94 to 1.05 | 0.823 |
| Missing smoking status | 1225 | 10 260 | 0.82 | 0.77 to 0.88 | <0.001 |
| IMD quintile |  |  |  |  |  |
| 1—least deprived | 2564 | 18 155 | 1.00 | – | – |
| 2 | 2490 | 17 635 | 0.94 | 0.88 to 1.00 | 0.054 |
| 3 | 2511 | 14 243 | 1.20 | 1.12 to 1.29 | <0.001 |
| 4 | 2413 | 12 859 | 1.13 | 1.06 to 1.21 | <0.001 |
| 5—most deprived | 2277 | 10 337 | 1.24 | 1.15 to 1.32 | <0.001 |
| Missing IMD | 903 | 5026 | 1.04 | 0.95 to 1.13 | 0.395 |
| CHD | 1993 | 9669 | 1.24 | 1.16 to 1.31 | <0.001 |
| Stroke | 535 | 2603 | 1.09 | 0.98 to 1.21 | 0.116 |
| Type 2 diabetes | 4401 | 12 884 | 1.83 | 1.75 to 1.92 | <0.001 |
| Depression | 6385 | 31 573 | 1.33 | 1.28 to 1.39 | <0.001 |

*BMI group at baseline. Patients could change BMI category in the analysis so intervention may have been delivered when patients had changed BMI category.
BMI, body mass index; CHD, coronary heart disease; IMD, Index of Multiple Deprivation.

**Table 5** Use of weight management interventions in general practices

| | Minimum | 10th centile | 25th centile | Median | 75th centile | 90th centile | Maximum |
|---|---|---|---|---|---|---|---|
| Patients receiving any intervention (%) | 0 | 4 | 7 | 12 | 19 | 28 | 91 |
| Patients receiving advice (%) | 0 | 0 | 0 | 3 | 9 | 18 | 91 |
| Patients receiving a referral (%) | 0 | 0 | 1 | 3 | 7 | 13 | 50 |
| Patients receiving antiobesity drugs (%) | 0 | 0 | 2 | 4 | 6 | 9 | 33 |

Figures are percentiles among 491 GP practices representing the proportion of patients in the practice receiving interventions.
NB: different practices may occupy centiles for different measures.

might also indicate a lack of patient access to appropriate body weight management interventions in primary care due to a lack of clinician awareness or confidence in treating obesity. Guidelines on the management of obesity from the National Institute for Health and Care Excellence (NICE) [3] do not appear to have been successfully implemented into practice.

There was some evidence that body weight management was tailored to obesity category with more frequent utilisation of antiobesity drugs in patients who were in higher obesity categories and advice used more commonly in overweight patients. While BMI category was the strongest predictor of a patient receiving weight management interventions, with rates over three times higher in morbid obesity than in overweight, female gender, increasing age, socioeconomic deprivation and comorbidities tended to be associated with greater use of weight management interventions.

Follow-up values for body weight after a recorded weight management intervention were limited. Monitoring of body weight in primary care is generally opportunistic and depends on patients attending the practice and having a weight measurement recorded. However, the relatively high levels of comorbidity in patients in this cohort, including those that require long-term management such as type 2 diabetes, suggest that consultations are likely to be regular. While follow-up weight measurements did not show any change in weight after intervention, these results are very vulnerable to information bias.

### Comparison with the literature

One other UK-based study investigated using of primary care interventions for the treatment of obesity.[9] The Counterweight report identified that 20% of patients received advice, 4% referrals and 2% antiobesity drugs based on a review of 100 obese patients medical records over an 18-month period in 2000–2001. We identified a smaller proportion of patients receiving advice and a higher proportion having a referral or drug prescription over a longer time period. It was not clear how obese participants were selected in the Counterweight study. Other differences between the present study and the Counterweight paper include a larger sample size and inclusion of overweight patients. However, the results suggest that prescribing of antiobesity drugs has increased in the past 15 years. Increased use of

antiobesity drugs between 1998, when they were first introduced in the UK, and 2005 has been reported elsewhere.[16]

A decline in lifestyle advice and counselling for weight loss given to obese patients over the past 10 years has been also been reported in studies from the USA. Reasons behind this reduction, despite increasing obesity levels, include poor recording of advice, lack of time in consultations, pessimism regarding potential success of weight loss attempts and increased use of medications to treat obesity-related risk factors and disease [17 18] and, perhaps, normalisation of excessive body weight. Although the evidence from the current study is not sufficient to conclude that a reduction in advice for weight management has occurred, some of the explanations attributed to lowered rates in the USA are likely to be applicable in the UK.

### Strengths and limitations

This study design had the advantage of a large population-based sample taken from different regions of the UK. However, it is likely that not all weight management interventions, particularly lifestyle advice, were captured in the electronic health record. Brief advice may be given to patients but not recorded by clinicians, which could have led to an underestimation of intervention rates. This is less likely to be an issue with referrals and drug prescribing. Furthermore, the patients included in this sample were selected on the basis of having a BMI record indicating that they were overweight or obese. This may have introduced a selection bias as these patients have been identified as having a weight problem by a clinician. Patients who are obese but do not have a record of weight status in their medical record may or may not be receiving weight management interventions differently from those who have been diagnosed.

### Implications for practice and future research

The results of this study suggest that primary care interventions given to patients with the aim of reducing weight are underutilised, and that follow-up to determine their success is poor. It is possible that rates have been underestimated through a lack of formal recording in medical records. However, the growing burden of obesity on primary healthcare services and lack of long-term follow-up on the effectiveness of these treatments supports the use of structured recording of interventions

for weight management and subsequent follow-up. This is particularly true given the heterogeneity of results from weight loss studies included in reviews of the effectiveness of primary care interventions for obesity and the need for further evidence specific to patient subgroups, for example, those with comorbidities.[8 19 20] Primary care referrals to commercial weight loss programmes have been found to be effective in trials.[21 22] Although this type of referral was not included in the present study, an analysis using primary care data could be valuable. Data in CPRD are not specific enough to permit this at present. Consistency of public health messages on the health risks associated with obesity should be promoted in primary care where clinicians have the opportunity to reach a large number of patients and utilise preventive as well as reactive treatment strategies.

**Contributors** HPB and MCG designed the study. ATP and MCG advised on the conduct of the data analysis. HPB conducted the analysis and drafted the paper. MCG and ATP contributed to interpretation of the results. All authors commented and approved the manuscript. HPB is the guarantor.

**Funding** This study was supported by the UK National Prevention Research Initiative whose funding partners include the Alzheimer's Research Trust; Alzheimer's Society; Biotechnology and Biological Sciences Research Council; British Heart Foundation; Cancer Research UK; Chief Scientist Office, Scottish Government Health Directorate; Department of Health; Diabetes UK; Economic and Social Research Council; Engineering and Physical Sciences Research Council; Health & Social Care Research & Development Office for Northern Ireland; Medical Research Council; The Stroke Association; Welsh Assembly Government; and World Cancer Research Fund. The views expressed are those of the authors alone.

**Competing interests** None.

**Ethics approval** CPRD Independent Scientific Advisory Committee (ISAC 07_054 and 14_056).

**Provenance and peer review** Not commissioned; externally peer reviewed.

**Data sharing statement** No additional data are available.

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
