## [Reviewer comments · BMJ Open]

ARTICLE DETAILS

TITLE (PROVISIONAL)	Access to weight reduction interventions for overweight and obese patients in UK primary care. Population-based cohort study
AUTHORS	Booth, Helen; Prevost, A.; Gulliford, Martin

VERSION 1 - REVIEW

REVIEWER	Tania Markovic Royal Prince Alfred Hospital Sydney Australia
REVIEW RETURNED	04-Oct-2014

GENERAL COMMENTS	This is an interesting paper and if it promotes discussion around the poor intervention for overweight in primary care, this alone will be a worthwhile result. I would be interested to know more about the patient selection, ie. how many of the total cohort (300,000) were ineligible just because there was no weight or height measurement? My guess is that this number is high and again this is worth mentioning. How often was BP measured?
--

REVIEWER	Dr Nicholas Fuller The Boden Institute, Sydney Medical School, The University of Sydney, Australia
REVIEW RETURNED	08-Oct-2014

GENERAL COMMENTS	This is a well written paper. The introduction clearly explains context of the research and the need to carry out such a study. Major points to consider: Overall conclusions of the paper need rethinking or at least touching on some other areas. Surely it is not a lack of access to appropriate body weight management interventions in primary care?? Despite only recently being implemented the UK has the NICE guidelines pathways – perhaps better education on appropriate referral pathways needs to be targeted. This needs mentioning. Abstract – Article Summary: Please rethink. One of the biggest limitations is that you haven't included commercial programme referrals. Why not? Especially considering they are a part of the NICE referral pathway. Analysis Page 8 Lines 5 – Explain what the percentiles refer to in Table 5. This part of your analysis is not discussed well in the methodology. Results Page 9 Lines 13- Considering rewording the latter half of this paragraph – the
--

	descriptive text of the patient characteristics is not clear. What is the diagnostic code? Mention in methods. Table 1. Must be stand alone. Spell out acronyms. Are the figures frequencies (column percent) or Mean (SD)? Table 3. These interventions need more detailed and clearer explanation of what they encompass in the methodology section e.g. "Advice, referral or obesity drug prescription". Page 13. Line 8. Please reword the text relating to predictors. Don't just list every variable. This section needs to be more concise. Table 4. Tables must be stand alone. Spell out acronyms. What does the P value refer to? Table 5. This table is not clear. Refer to previous comment. What do the percentiles refer to? Please change the layout for better clarity. Discussion – as mentioned earlier. Page 15. Line 26. Really? Or is it a lack of education and awareness by the GPs/ primary care professionals? Page 16. Line 34. Suggest removing this sentence. Rates of recorded advice may have fallen? What is this assumption based on? Page 17. Line 57. You introduce a new idea into the paper. Discuss this earlier in results – What are you basing the mixed results of the effectiveness of primary care interventions for obesity on? They are consistent. Eg. Two of the larger studies - Counterweight and Jebb et al (WW paper) both have approx 2-3 kg weight loss. Page 18. Line 7. As per comment above, please elaborate on why commercial weight loss programme referral was not included – this is a large limitation of the paper. STROBE checklist – You have not addressed point 13. Participants. Minor points: Correct spelling for WHO is World Health Organization. Make sure all acronyms are spelt out the first time e.g. CHD.
--	---

VERSION 1 – AUTHOR RESPONSE

Reviewer: 1

This is an interesting paper and if it promotes discussion around the poor intervention for overweight in primary care, this alone will be a worthwhile result. I would be interested to know more about the patient selection, ie. how many of the total cohort (300,000) were ineligible just because there was no weight or height measurement? My guess is that this number is high and again this is worth mentioning. How often was BP measured?

Thank you for your comment. We are pleased you feel the paper will promote interest in this topic. The cohort was selected as part of a larger project, which we have now acknowledged in the methods on p.6 "The initial cohort was selected as part of a larger project, and comprised a random sample of approximately 300,000 patients..." In total, 45% of patients had a BMI recorded during the study period. We have added a sentence to the results (p.9) to clarify this "Of the 300,006 patients in the cohort, 134,697 (45%) had an eligible BMI record. After patients with BMIs lower than 25kg/m² were removed, data were analysed for 91,413 patients". The discussion already states that patients who are not recorded as obese may be receiving weight management interventions differently from those with diagnosed obesity (p.17).

We did not elect to investigate measurement of blood pressure and other risk factors in this paper as this is a complex topic in its own right.

Reviewer: 2

This is a well written paper. The introduction clearly explains context of the research and the need to carry out such a study.

Thank you for this feedback.

Major points to consider:

Overall conclusions of the paper need rethinking or at least touching on some other areas. Surely it is not a lack of access to appropriate body weight management interventions in primary care?? Despite only recently being implemented the UK has the NICE guidelines pathways – perhaps better education on appropriate referral pathways needs to be targeted. This needs mentioning.

Thank you for your comment. The NICE guidelines provide recommendations for the management of overweight and obesity, but these recommendations have not been implemented into practice, as evidenced by these results. We have amended the abstract to read “may indicate a lack of patient access to appropriate body weight management interventions in primary care” to highlight that the interventions are available but possibly not accessible to patients due to them not necessarily being offered by clinicians.

Abstract – Article Summary:

Please rethink. One of the biggest limitations is that you haven't included commercial programme referrals. Why not? Especially considering they are a part of the NICE referral pathway.

Thank you for your comment. Commercial weight loss programmes were not included in this analysis as public and private providers are not distinguished in the patient medical record at present. Medical codes only describe a referral to a weight management programme and do not provide information on the programme provider. As mentioned in the discussion (p.18) an evaluation of these programmes would be valuable using primary care data to extend RCT analyses. We have added the following sentence to p.18 “Data in CPRD are not specific enough to permit this at present.”

Analysis Page 8 Lines 5 –

Explain what the percentiles refer to in Table 5. This part of your analysis is not discussed well in the methodology.

Thank you for this suggestion. The title of table 5 has been updated, and the methods amended to read: “Variation in the use of weight management interventions by GP practice was investigated by calculating the proportion of patients receiving any intervention in the year following the index date. These data were then presented as percentiles of the distribution for all practices.” (p.8).

Results Page 9 Lines 13-

Considering rewording the latter half of this paragraph – the descriptive text of the patient characteristics is not clear. What is the diagnostic code? Mention in methods.

Thank you for your comment. We have reworded this paragraph on p.9 as follows: “. At the index date (date of the first relevant BMI record) most patients were overweight (63.9% of men and 56.2% of women); 2.9% of men and 6.3% of women were morbidly obese. A diagnostic code for obesity was recorded for 3.9% of male patients and 6.5% of females.” The methods have been updated to clarify diagnostic coding on p.6 “Medical diagnoses of obesity in the medical record were also noted based on the presence of diagnostic codes.”

Table 1. Must be stand alone. Spell out acronyms. Are the figures frequencies (column percent) or Mean (SD)?

Thank you for this comment. We have updated the title to read "Figures are frequencies (column percent) unless stated otherwise." SD is only presented in relation to age and is stated as such. All acronyms have been explained.

Table 3. These interventions need more detailed and clearer explanation of what they encompass in the methodology section e.g. "Advice, referral or obesity drug prescription".

Thank you for this comment. We have revised the relevant paragraph on p.7 including the following statement: "Advice included codes relating to dieting, exercise and weight loss. Relevant referrals included those to community and hospital dieticians, for exercise therapy and for weight management programmes." Unfortunately the medical codes are limited in their description of what's offered, for instance medical code 26276 describes "Referred to exercise programme" so the available information is limited.

Page 13. Line 8. Please reword the text relating to predictors. Don't just list every variable. This section needs to be more concise.

Thank you for your suggestions. We have not revised the text as these variables were not the main focus of the paper and the BMJ Open manuscript preparation guidelines state that data in the tables should not be duplicated in the text.

Table 4. Tables must be stand alone. Spell out acronyms.
What does the P value refer to?

Thank you for your comment. We have updated this table accordingly. The P value relates to the accuracy of the hazard ratio estimate.

Table 5. This table is not clear. Refer to previous comment. What do the percentiles refer to? Please change the layout for better clarity.

Thank you for your comment. We have updated the title of this table as described above. We produced the table layout in line with the journal requirements so have not reformatted it at this stage.

Discussion – as mentioned earlier. Page 15. Line 26. Really? Or is it a lack of education and awareness by the GPs/ primary care professionals?

Thank you for your comment. The last sentence of the first discussion paragraph of p.15 has been amended as follows: "but might also indicate a lack of patient access to appropriate body weight management interventions in primary care due to a lack of clinician awareness or confidence in treating obesity."

Page 16. Line 34. Suggest removing this sentence. Rates of recorded advice may have fallen? What is this assumption based on?

Thank you for this suggestion. We have removed the relevant sentence.

Page 17. Line 57. You introduce a new idea into the paper. Discuss this earlier in results – What are you basing the mixed results of the effectiveness of primary care interventions for obesity on? They are consistent. Eg. Two of the larger studies - Counterweight and Jebb et al (WW paper) both have

approx 2-3 kg weight loss.

Thank you for your comment. We have amended the sentence on p.17 to read “This is particularly true given the heterogeneity of results from weight loss studies included in reviews of the effectiveness of primary-care interventions for obesity”. The systematic reviews referred to in that sentence found weight loss to be highly heterogeneous. This sentence does not refer to the commercial weight loss programmes discussed afterwards. We have not amended the results as this point relates to the interpretation of our findings rather than the analysis itself.

Page 18. Line 7. As per comment above, please elaborate on why commercial weight loss programme referral was not included – this is a large limitation of the paper.

Thank you for your point. As discussed above this isn't possible using CPRD electronic health records and the following sentence has been added to p.18 “Data in CPRD are not specific enough to permit this at present.”

STROBE checklist – You have not addressed point 13. Participants.

Thank you for your comment. We have updated p.9 to read: “Of the 300,006 patients in the cohort, 134,697 (45%) had an eligible BMI record. After patients with BMIs lower than 25kg/m² were removed, data were analysed for 91,413 patients”

Minor points:

Correct spelling for WHO is World Health Organization.

Make sure all acronyms are spelt out the first time e.g. CHD.

VERSION 2 – REVIEW

REVIEWER	Tania Markovic Royal Prince Alfred Hospital Australia
REVIEW RETURNED	01-Nov-2014

GENERAL COMMENTS	I am happy with the changes and recommend this paper be published.
--

REVIEWER	Dr Nicholas Fuller The Boden Institute, Sydney Medical School, The University of Sydney
REVIEW RETURNED	10-Nov-2014

GENERAL COMMENTS	The manuscript reads much clearer now. Thank you. Some minor revisions: It would be beneficial to add to the manuscript that the NICE guidelines do not appear to be being implemented into practice as evidenced by your results. Page 13. Line 8. Please reword the text relating to predictors. Don't just list every variable. This section needs to be more concise. ...This comment referred to the fact that perhaps only the strongest predictor should be described in the text – because as you say, these variables were not the focus of this paper.
--

VERSION 2 – AUTHOR RESPONSE

Reviewer 2.

Comment 1: We now say (page 15) 'Guidelines on the management of obesity from NICE (3) do not appear to have been successfully implemented into practice.'

Comment 2: We have changed the wording to read (page 13) 'Increasing age, type 2 diabetes and depression tended to be associated with receiving a weight loss intervention. ' and page 15 ' While BMI category was the strongest predictor of a patient receiving weight management interventions, with rates over 3-times higher in morbid obesity than in overweight, female gender, increasing age, socioeconomic deprivation and co-morbidities tended to be associated with greater use of weight management interventions.'

We hope these changes meet your requirements. Thank you for considering this resubmission of the paper.